# Disseminated Intravascular Coagulation (DIC): Old player creates new perspectives on the polymicrobial sepsis model of CASP

**Julia van der Linde**[1]*, **Stephan Diedrich**[1], **Thorben Klee**[2], **Claus-Dieter Heidecke**[1], **Stephan Kersting**[1], **Wolfram Keßler**[1]

**1** Department of General, Visceral, Thoracic and Vascular Surgery, University Medical Center Greifswald, Greifswald, Germany, **2** Department of Anesthesiology and Intensive Care Medicine, University Medical Center Schleswig-Holstein, Lübeck, Germany

* Julia.vanderlinde@med.uni-greifswald.de

## Abstract

**Data Availability Statement:** All relevant data are within the paper and its Supporting Information files.

### Background

Disseminated Intravascular Coagulation (DIC) is a life-threatening complication of sepsis. In surgical ICUs, DIC is frequently caused by abdominal sepsis, and the disarranged coagulation and complications often lead to death. The severity of sepsis is associated with a higher DIC score according to the parameters proposed by the International Society of Hemostasis and Thrombosis (ISTH) in 2001: platelet count, bleeding time (Quick), D-dimer, and fibrinogen. One problem in studying DIC is finding an adequate animal model that reflects the clinical situation of polymicrobial overwhelming infection.

### Aims and methods

We investigated whether a well-established polymicrobial sepsis model of colon ascendens stent peritonitis (CASP) is suited to investigate the complexity of DIC. For this purpose, CASP-operated mice were examined 20 h after the operation with regard to coagulation parameters using cell counts, bleeding times, rotational thromboelastometry (ROTEM), ELISAs for D-dimer and fibrinogen, and platelet accumulation in affected organs via immunohistochemistry to see if the mice develop a coagulation disorder that meets the definition of DIC proposed by the ISTH 2001 consensus conference.

### Results

Herein, we showed that the CASP model is an all-encompassing animal model to analyze the complexity of systemic DIC in murine abdominal sepsis. There is highly reproducible thrombocytopenia, a significant prolongation of the bleeding time, and a loss of fibrinogen in plasma. We also observed microvascular thrombosis due to platelet accumulation in the microcirculation of the liver.

**Funding:** This work was supported by the German Research Foundation in the form of a grant to JvdL [project no. 273551 GRK 840: Host Pathogen Interactions in Generalised Bacterial Infections].

**Competing interests:** NO - The authors have declared that no competing interests exist.

## Conclusion

The CASP model seems superior to other artificial models, e.g., injecting substances, for inducing DIC. CASP is one of the best true-to-life models for analyzing the complexity of disseminated intravascular coagulation in polymicrobial sepsis.

## Introduction

More than ever, the significance of a misdirected immune response in the context of sepsis with additional disarranged coagulation is relevant. Disseminated intravascular coagulation (DIC), caused by bacterial hyperinflammation, is still not completely understood. New scoring systems and additional criteria to define DIC are currently under evaluation [1, 2]. The complexity and necessity of understanding coagulation disorders in the critically ill was highlighted during the COVID-19 pandemic [3], where the influence of coagulation abnormalities on the prognosis of critically ill patients was illustrated. This observation illustrates the significance of coagulation's influence on the prognosis of critically ill patients.

Disseminated intravascular coagulopathy is a life-threatening complication of sepsis. Aside from severe sepsis, other diseases can cause DIC, such as malignancy, trauma, liver cirrhosis, snake bites and obstetric complications. However, sepsis is one of the most common causes of DIC, with DIC being one of the dominant influential factors affecting its outcome [4]. The presence of DIC doubles the sepsis mortality rate. During the body's response to the underlying diseases, there are phases of increased and decreased coagulation characterized by microthrombotic deposits in the capillary flow area or intraparenchymal hemorrhaging in the organ. Thrombotic deposits lead to microinfarction and reduced oxygen supply to the organ, subsequently followed by organ failure with septic shock [5]. The pathologic significance of DIC is the simultaneous occurrence of hypercoagulation and hypocoagulation. Developing fibrin deposits increase the consumption of platelets and coagulation factors, and the simultaneous presence of plasmin causes hyperfibrinolysis [6].

In 2001, a uniform DIC assessment was drawn up at the consensus conference of the International Society on Thrombosis and Hemostasis (ISTH). The platelet count, bleeding time in the form of the prothrombin index in %, D-dimer level, and fibrinogen level are among the most important parameters of the ISTH score [7]. DIC is also often present in sepsis with multiple organ failure.

Challenges to developing treatments are its currently unclear basic etiology and the role of its interactions with the immune system. Existing animal DIC models can be categorized according to medication-induced (tissue factor, thrombin or snake poison) and infectious variants, such as DIC caused by LPS (lipopolysaccharide) or bacteria. None of these models present a suitable variant that takes into account the complexity of a polymicrobial infection with a reaction by the immune system. Regarding DIC, the effect of the inflammasome on complex blood components, such as platelets, plays a vital role. Recent data imply that the activation of the inflammasome is the decisive factor in mortality [8]. The inflammasome introduced by Jurg Tschopp and colleagues in 2002, is a term explaining a multi protein complex belonging to the innate immune system. Triggered by intracellular NOD-like-receptors (NLRs), detecting danger signals in euraryotic cells of bacterial origin, it consists of NLR-protein (a family of 22 NLR), an Adapter protein (ASC) and inflammatory Caspases, e.g. Caspase 1. The inflammatory Caspases where activated by intracellular bacterial products e.g. Flagellin or LPS leading to Pyrocytosis, a form of lytic cell death, causing the release of Interleukin 1beta and

Interleukin 18 [9, 10]. In previous studies, our working group was able to demonstrate a strong increase in interleukin 1 beta in the CASP model [11]. In Pyrocytosis the plasmamembrane rapidly ruptures and thrombogenic substances as Tissue Factor are released, which additionally pours coals into the fire of DIC and triggers coagulation cascade [8, 12]. Beside this the inflammasome is currently discussed as misguided immune responses in chronic inflammatory situations, metabolic syndrome, gout and as a reason for islet cell death in type 2 diabetes in literature.

The current literature criticizes the existing artificial DIC models for not fully incorporating the complexity of sepsis [13].

The model of colon ascendens stent peritonitis (CASP) is an intra-abdominal sepsis model created by hollow organ perforation. Pursuant to the S3 consensus conference, human sepsis is defined as a life-threatening situation due to a dysregulated organ response to infection [14]. The CASP model seems to reflect the organ dysfunction observed in human sepsis, while other animal models are appropriate to reproduce endotoxin shock (LPS model) or to display systemic inflammatory response syndrome (SIRS), such as cecal ligation and puncture (CLP). In these models, dealing with bacterial pathogens, Bacteremia is vital for developing DIC, which is produced in the CASP model by continuous leakage in the area of the cecum.

In this paper, we present our comprehensive CASP coagulation analyses and based on our findings, we recommend using the CASP model for the complex analysis of infectious DIC in animal models because CASP, with a repetitious accuracy, is able to recreate a realistic interaction of the coagulation parameters in the complex situation of severe polymicrobial sepsis.

## Materials and methods

All experiments were performed on 8- to 10-week-old female C57BL/6 mice. The animals were housed in a temperature-controlled room with a 12-hour light and dark cycle. All experiments were approved by the animal ethics committee of the local animal protection authority (LALLF, State Office for Agriculture, Food Safety and Fisheries Mecklenburg-Western Pomerania; LALLF M-V/TSD/7221.3.-1.1-056/12 and 7221.3-1-056/14). The results shown were created as part of the animal experiments of these projects. In total, the data from 117 C57Bl/6 mice are shown. Different numbers of animals in the groups are originated by our efforts to keep the control groups as small as possible, if the experimental setting allows us to expect stable results without a wide range of fluctuation. In addition, we considered that animals in the CASP group could be withdrawn due to excessive stress. In order to avoid small numbers in CASP group, more animals where included, to obtain meaningful results. All efforts were made to minimize the suffering of the mice, enrichments were provided.

### Colon ascendens stent peritonitis (CASP)

CASP surgery was carried out according to the method described by Traeger et al. 2010 [15]. A solution of 0.5 ml xylazine (20 mg/ml), 1.0 ml ketamine (100 mg/ml) and 8.5 ml isotonic NaCl solution was used to anesthetize the mice. By i.p. application of 10 μl anesthetic / g body weight, the animals reached a sufficient depth of anesthesia after about 5 minutes. This was checked by the disappearance of pain stimuli on the tail and the inability to trigger leg reflexes. About an hour after application, the animals became lively again and were able to move freely after 2–3 hours.

A 16 G cannula (BD Venflon, Franklin Lakes, USA) was introduced 15 mm distal to the ileocecal valve and attached with 7/0 sutures. This stent caused continuous leakage in the area of the cecum and simulated postoperative polymicrobial sepsis. The abdominal wall was closed after repositioning the intestine.

## Analgesia and monitoring

All mice received 0.1 mg/kg buprenorphine intramuscularly per day to treat pain. Stress scores were collected and evaluated every 6 hours. If the stress severity score (SSS) on the animals exceeded a score of 10, they were killed painlessly under anesthesia by cervical dislocation [16]. Some values of n in this study are occasionally caused by animals that were exceeded according to their stress severity score (SSS > 10).

## Measurement of the platelet count

The platelet count was measured in the whole blood of the model vs. the untreated animals. To do so, we used an impedance-based measurement method (VetScan®, Abaxis, Union City, USA). After 20 hours, the animals' blood was collected by retro-orbital puncture followed by anticoagulating with EDTA.

## Measurement of the bleeding time

To determine the bleeding time, we clipped off 2 mm of the tail tip with a scalpel and measured the time until the onset of hemostasis, following the method of Dejana et al. [17]. The bleeding was made visible by transferring the wounded region into a Falcon tube filled with 15 ml of 0.9% NaCl solution at 37˚C. If hemostasis did not occur within 300 sec, the measurement was stopped, and the value was set to 300 sec. The bleeding time was measured in 3 different experiments done within one month, 23 untreated animals were included and 25 CASP-operated animals, we used animals that where proceeded for further analysis, where the Tail-Tip-bleeding test would not interfere with results.

## Rotational thromboelastometry

Then, 400 µl of citrate-anticoagulated blood from each mouse was added to ex-Tem® S (tissue thromboplastin) corresponding to the manufacturer's specifications for determining the extrinsic coagulation cascade as a correlate for testing the prothrombin and clotting time (CT) using ROTEM® delta (TEM international GmbH, Munich, Germany). The ascertained values were plotted against each other. Because the analyzer can only evaluate 4 samples at the same time, data from two experimental days were collected to minimize the time between blood sampling and analysis of the probes in ROTEM® delta.

## ELISA

**D-dimer ELISA.** To analyze the D-dimer concentration of the septic and untreated mice, plasma at a 1:6 dilution was measured with the *Stago* Asserachrom® D-dimer ELISA kit (Asnieres sur Seine, France). All steps were carried out according to the manufacturer's specifications, measuring the OD after terminating the reaction on the ELISA reader at 450 nm. Standard curves and concentrations were prepared and analyzed using GraphPad Prism 8 software.

**Fibrinogen ELISA.** The enzyme-linked immunosorbent assay (ELISA, by the company *Abnova*, Taipei, Taiwan) was conducted with an anti-fibrinogen antibody precoated 96-well microtiter plate according to the manufacturer's specifications. The plate was incubated with diluted mouse plasma (1:30,000) for 10 minutes at room temperature. The plasma was produced from anticoagulated whole mouse blood by centrifugation for 15 minutes at 2,000 g. Reaction detergents were added according to the ELISA manufacturer's specifications. The optical density was measured using an ELISA microplate reader after terminating the reaction.

Standard curves and concentrations were prepared and analyzed using GraphPad Prism 8 software.

### Immunohistological staining

The organ cryosections were each incubated with 20 μl 20% FCS buffer for 10 minutes at room temperature and then treated with peroxidase blocking reagent for 10 minutes, followed by washing the sections twice with PBS. For single staining of platelets in the sections of the liver, anti-fibrinogen antibodies were not applied. The primary antibody anti-CD42c FITC (Emfret, Würzburg, Germany) was incubated at 5 μl in 100 μl 20% FCS buffer at 4˚C, the sections were washed again, and then incubated with the secondary antibody Alexa 488 at a dilution of 1:100.

We used 100 μl of the antibody mixture for the double staining of fibrinogen/CD42c (CD42c 1:100; fibrinogen 1:50 in PBS/20% FCS), which was incubated at 4˚C. Serving as an isotype control, the second organ section was treated with PBS/20% FCS (the isotype controls are not shown). Subsequently, the sections were incubated with 50 μl of the peroxidase-coupled secondary antibody (1:100). After additional washing, the organ sections were immersed in amplification diluent with 50 μl biotonyl tyramide (1:200), and after washing, 50 μl of anti-FITC-Alexa 488 (1:100) and streptavidin TRITC (1:200) were added in PBS/20% FCS. In the final step, the organ sections were treated with fluoroprep, and after examination with a Keyence Fluorescence Microscope BZ-9000 (Keyence, Osaka, Japan) at 20 x optic, all microscope images were evaluated using the BZ-II Analyzer. The TRITC positive area was analyzed with ImageJ software.

### Hematoxylin-eosin (H.E.) staining

The liver removed postmortem was embedded in paraffin-blocks, according to a standard protocol, sectioned into 3 μm slices on the microtome, fixed on a slide and stained. Sections were deparaffinized in two consecutive xylene baths for 5 min each. The sections were hydrated using a descending ethanol series (95%, 80%, 70%; 5 min each) and rinsed in double-distilled water. This was followed by incubation in Mayer's hematoxylin solution for 15 minutes before rinsing again with double distilled water. Subsequent counterstaining was performed with eosin Y and acetic acid for 4 min followed by rinsing twice. Dehydration was then carried out using an ascending ethanol series (70%, 80%, 90%) and cleaning with two xylene baths for 5 min each. The sections were covered with glycerine gelatin.

### Statistical analysis

Statistical analysis was calculated with GraphPad Prism version 8.0.0 for Windows, GraphPad Software, San Diego, California USA. When data are shown as significantly different, the p value is two-tailed $< 0.05$ by a Mann–Whitney test. The group sizes are mentioned in the Results section. All values are marked with dots. The median values are depicted as bold lines, and the interquartile Range (IQR) is depicted and mentioned in the legend of the figures (in brackets).

## Results

### Platelet count

The platelet count in the septic animals dropped very significantly to values well below the normal range (Fig 1). The experiments were carried out repeatedly, and all mice developed thrombocytopenia 20 hours after CASP induction. We were able to show a significant decrease in

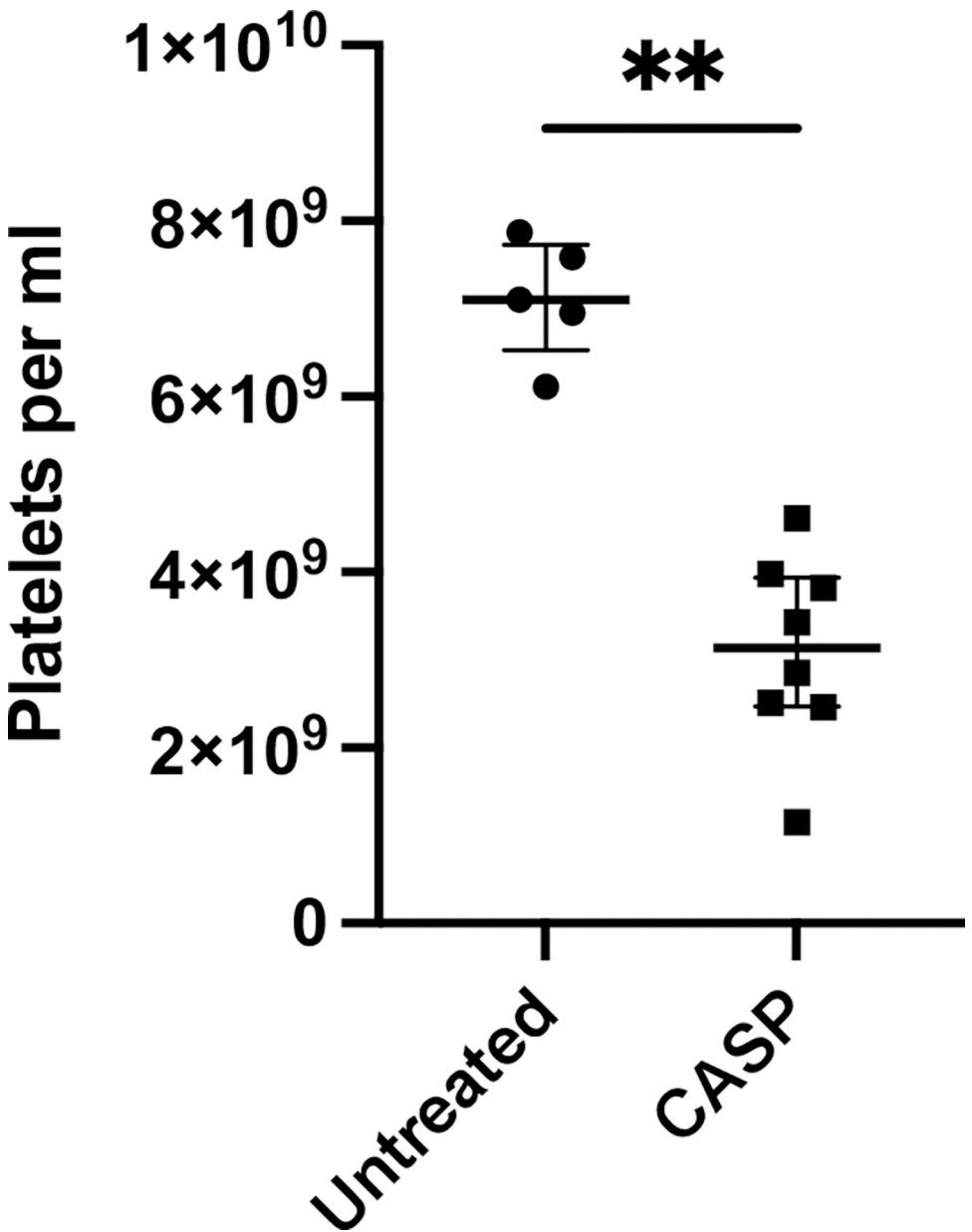

**Fig 1. Platelet count 20 hours after CASP surgery.** Untreated mice had a mean platelet count of 7.1 (1.76 IQR) $\times 10^9$/ml, white column. Mice that underwent CASP surgery showed a reduction to 3.14 (3.46 IQR) $\times 10^9$/ml, black column. Septic mice lost 60% of their platelets. The decrease in platelet count was highly significant (p = 0,0016**; n = 5 untreated group; n = 8 CASP group). The bold black line depicts the median.

the platelet count to 370.000 platelets/μl below the reference value of mouse C57Bl/6 (450.000–900.000 platelets/μl) reproducibly occurred 20 hours after inducing sepsis. Counts below 450.000 platelets/μl were consistently reproducible after CASP surgery, decreasing from 7.1 (1.76 IQR interquartile range IQR) $\times 10^9$ platelets/ml to 3.14 (3.46 IQR) $\times 10^9$ platelets/ml. On average, the model mice lost approximately 60% of their circulating platelets (p = 0.0016; n = 5 untreated group; n = 8 CASP group).

## Tail tip bleeding time

We analyzed the bleeding time of septic and untreated mouse tail tips. Septic mice showed a significantly prolonged bleeding time compared to the untreated control animals (Fig 2). On average, the median bleeding time of septic animal tail tips was 93 (260 IQR) seconds. In contrast, untreated animals showed a bleeding time of 57 (197 IQR) seconds. This difference was highly significant (p < 0.001; n = 23 untreated group; n = 25 CASP group).

## Thromboelastometry

**Clotting time (CT)–enzyme activity.** Rotational thromboelastometry (ROTEM) was used to obtain a correlate of the prothrombin time test according to the specifications of the ISTH 2001 DIC criteria. Clotting time (CT) was determined using EXTEM and clot formation time and MCF were also measured (data are not shown due to their similar results to CT). After 20 hours of CASP induction, we detected a prolonged median CT in the model mice of 109 (409 IQR) seconds, while the untreated control animals showed a CT of 80 (21 IQR) seconds. The variances were significantly different. The results are depicted in Fig 3. This difference also proved to be highly reproducible and the p value was p = 0.0043 (n = 6 untreated group; n = 5 CASP group).

**D-dimer.** D-dimer levels were also calculated using ELISA. Not surprisingly, 20 h after CASP surgery, there was a significant increase in the blood levels of D-dimer (Fig 4). The basal D-dimer levels of the untreated animals showed median values of approximately 120.9 (66.2 IQR) ng/ml, while 20 h after CASP surgery, it increased to 278.6 (722.9 IQR) ng/ml (p < 0.0001; n = 8 untreated group; n = 11 CASP group).

**Fibrinogen.** Fibrinogen analysis in the plasma of septic mice was carried out by comparing septic with untreated mice using a sandwich ELISA technique, which revealed a significant decrease (CASP: 29.71 (126.3 IQR) mg/dl blood plasma vs. untreated: 185.2 (83.91 IQR) mg/dl blood plasma) (p = 0.03, n = 4 untreated group, n = 5 CASP group). The results are depicted in Fig 5. These results could be verified by morphological imaging via immunohistochemical staining (Fig 6).

**Platelet deposits in the liver.** To demonstrate thrombus formation in shock organs, we conducted HE staining of the liver (spleen and lung but could not find any thrombus formation). To answer the question regarding the fate of the missing platelets, we decided to proceed with further platelet-specific staining of the respective organs and found significant platelet accumulation in the microvascular bed of the liver, as shown in Fig 7. Here, we present representative results and summarize the results in Fig 8. We could clearly demonstrate a significant increase in the accumulation of platelets in the liver.

Compared to the untreated control group (p = 0.0002), the liver of C57Bl/6 mice showed a significant increase in the platelet-specific marker CD42c (green dye Fig 8). On average, the control mice had a FITC-positive area of 0.54% (0.6% IQR) (n = 10). The medium fluorescence signal in the CASP-operated animals increased to 1.77% (1.69% IQR) of the area (n = 7). We could not find any differences in spleen and lung.

The double staining of fibrinogen and platelets in CASP sepsis clearly verified increased fibrinogen in the area of the sinusoids in untreated mice, while fibrinogen was less readily seen in those with sepsis.

## Discussion

DIC is a very complex and dynamic process. The different routine test methods continuously show a point-in-time snapshot of the coagulation situation with coexisting increased coagulation and hyperfibrinolysis. To understand the complexity of this process, it is reasonable to

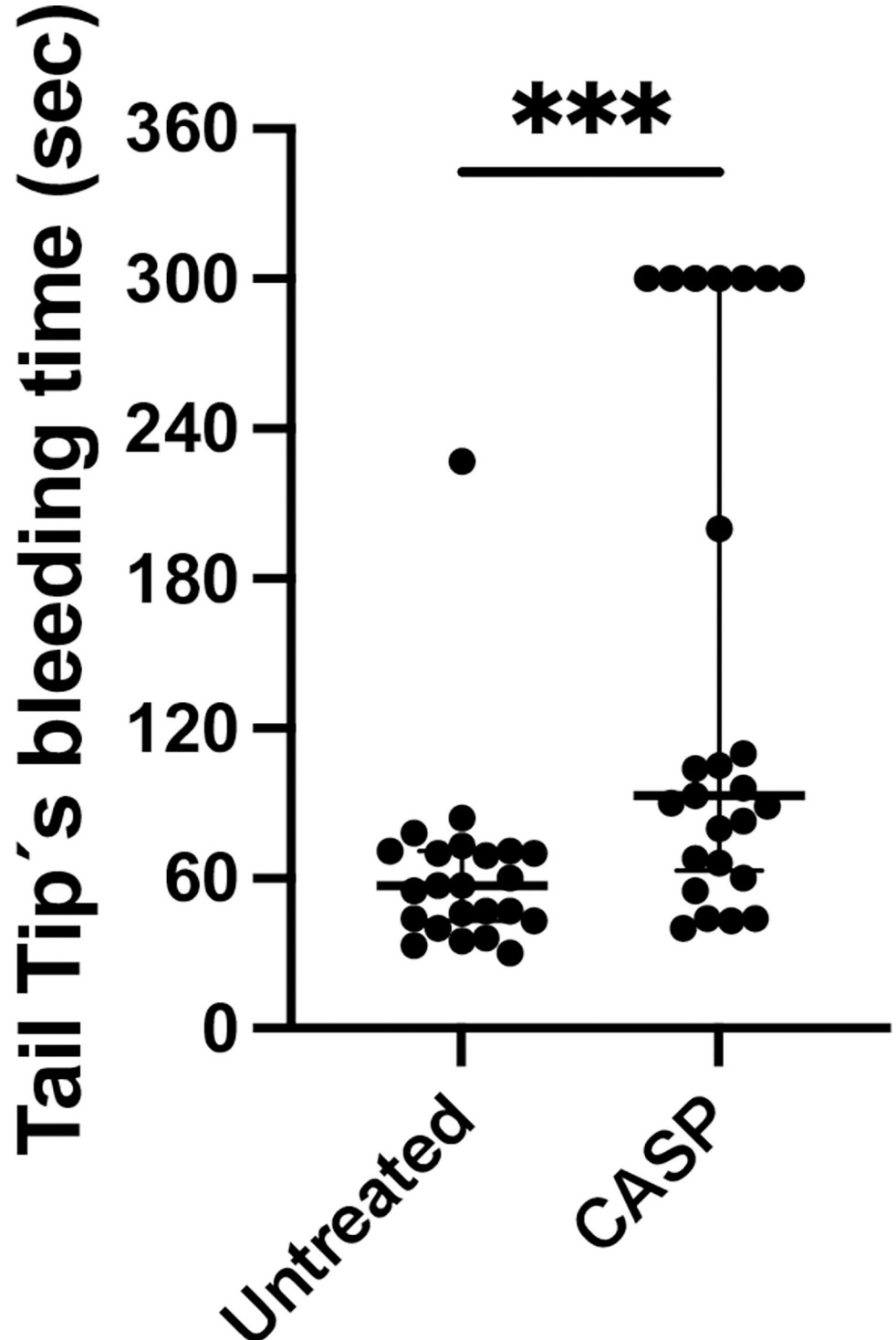

**Fig 2. Tail tip bleeding time in seconds.** Shown are bleeding times out of the tail tip of 8- to 10-week-old C57Bl/6 mice divided into CASP-operated, black column, and untreated animals, white column, 20 hours after CASP induction exhibiting significant differences in bleeding time (p < 0.001***; n = 23 untreated group; n = 25 CASP group). The tail tip bleeding time was 93 (260 IQR) seconds in septic mice and untreated mice showed a bleeding time of 57 (197 IQR) seconds. The bold black line depicts the median.

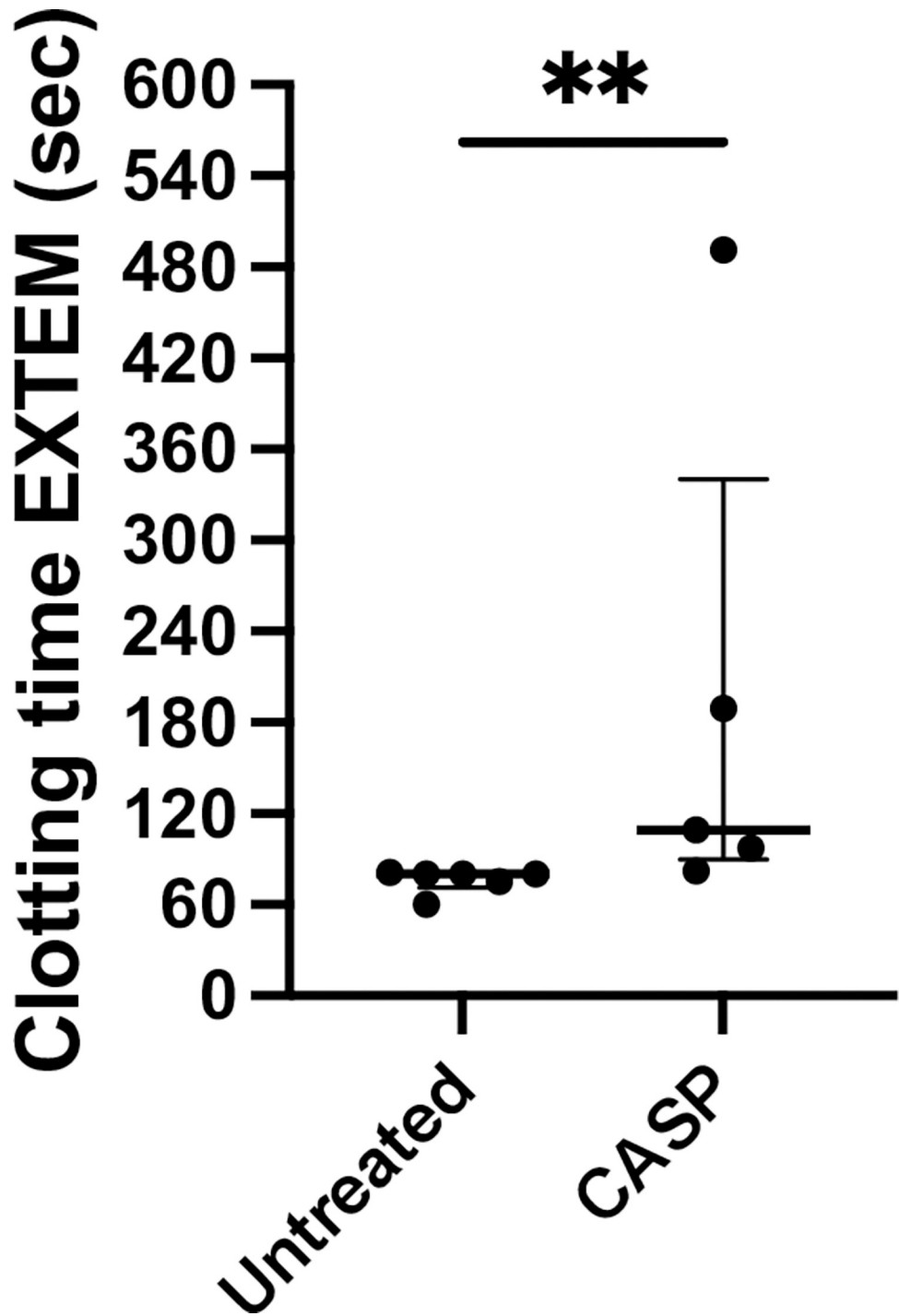

**Fig 3. Clotting time (CT) extrinsic pathway rotational thromboelastometry ROTEM I.** Illustrated is the CT, measured via rotational thromboelastometry EXTEM in seconds of female 8- to 10-week-old C57Bl/6 mice untreated (white column) or 20 h after CASP surgery (CASP, black column); control mice (untreated) exhibited a stable clotting mechanism within 80 (21 IQR) seconds while 20 h after CASP surgery, the mice showed a prolongation of CT to 109 (409 IQR) seconds (p = 0.0043**; n = 6 untreated group; n = 5 CASP group). The bold black line depicts the median.

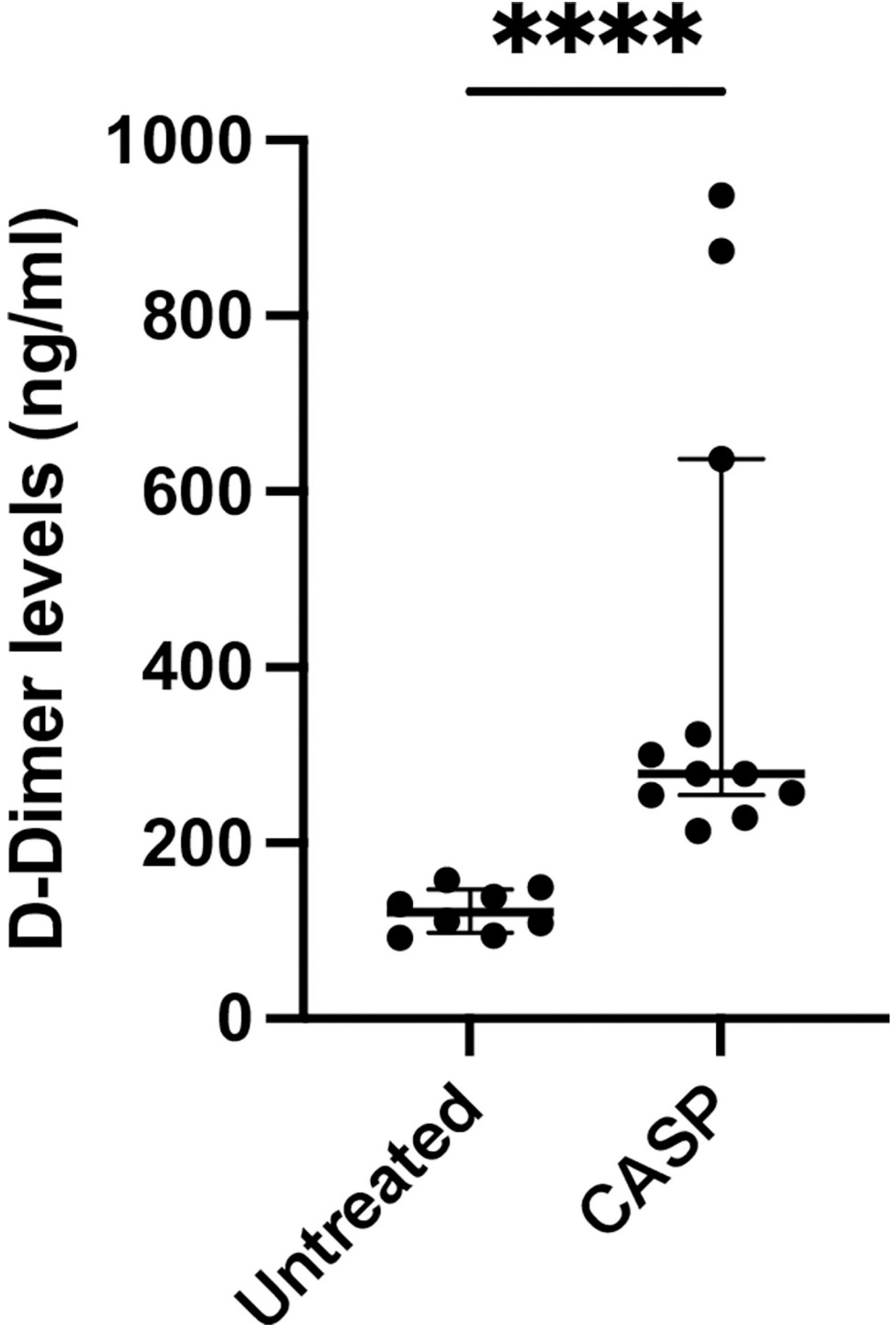

**Fig 4. D-dimer levels in the blood of untreated or CASP-operated C57Bl/6 mice.** Statistical results of D-dimer levels 20 h after CASP surgery (CASP, black column) or untreated (white column) 8- to 10-week-old C57Bl/6. Basal blood levels of untreated animals 120.9 (66.2 IQR) ng/ml increased significantly to 278.6 (722.9 IQR) ng/ml in CASP-operated mice (p < 0.0001****; n = 8 untreated group; n = 11 CASP group); The bold black line depicts the median.

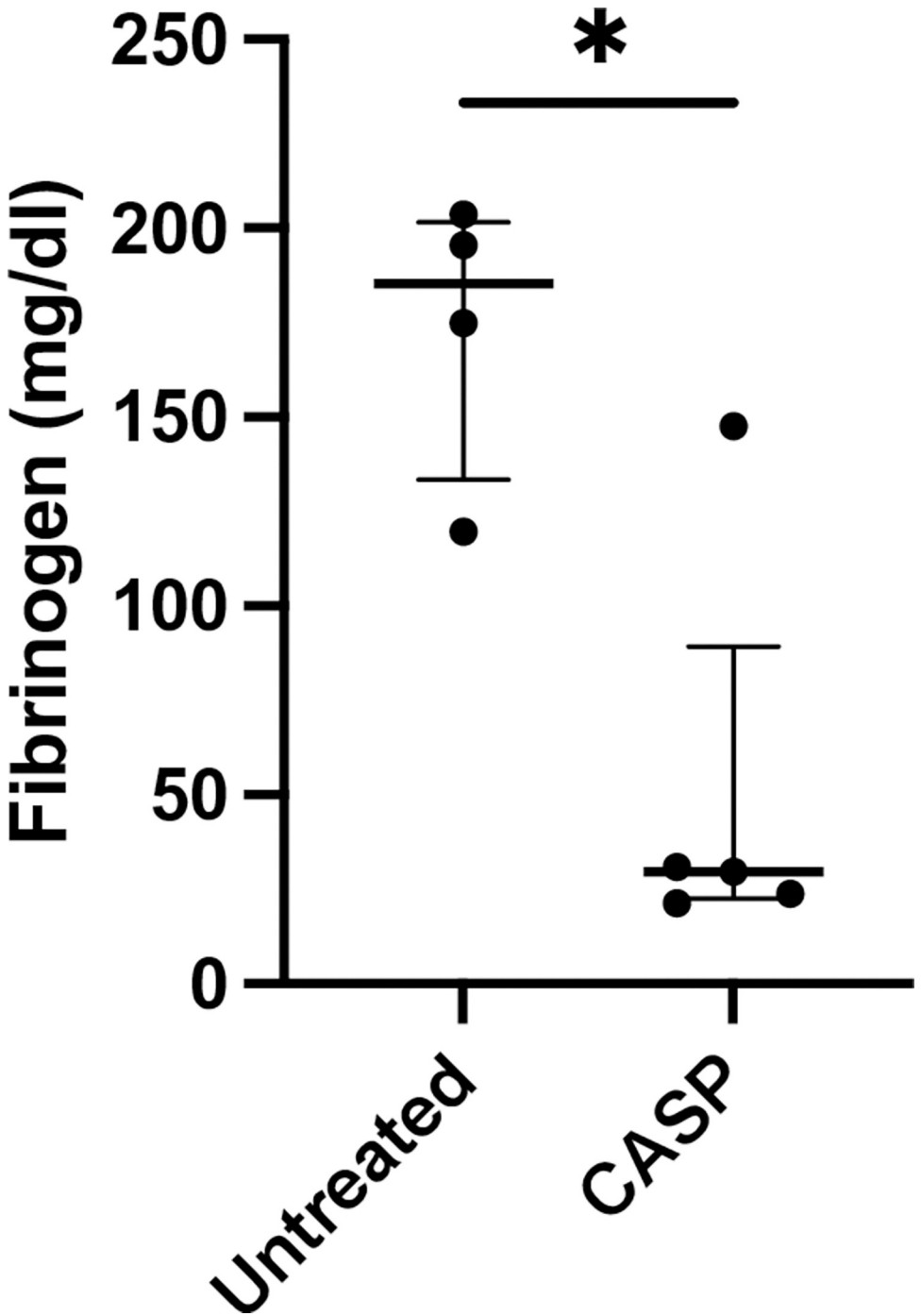

**Fig 5. Fibrinogen blood levels 20 h after CASP surgery.** Statistical results of fibrinogen blood levels 20 h after CASP surgery in 8- to 10-week-old C57BL/6 mice depicting a significant decrease in fibrinogen plasma levels in CASP-operated mice (CASP, black column). Control animals (untreated, white column) showed 185.2 (83.91 IQR) mg/dl vs. 29.71 (126.3 IQR) mg/dl blood levels 20 h after CASP surgery (p = 0.03*; n = 4 untreated group; n = 5 CASP group). The bold black line depicts the median.

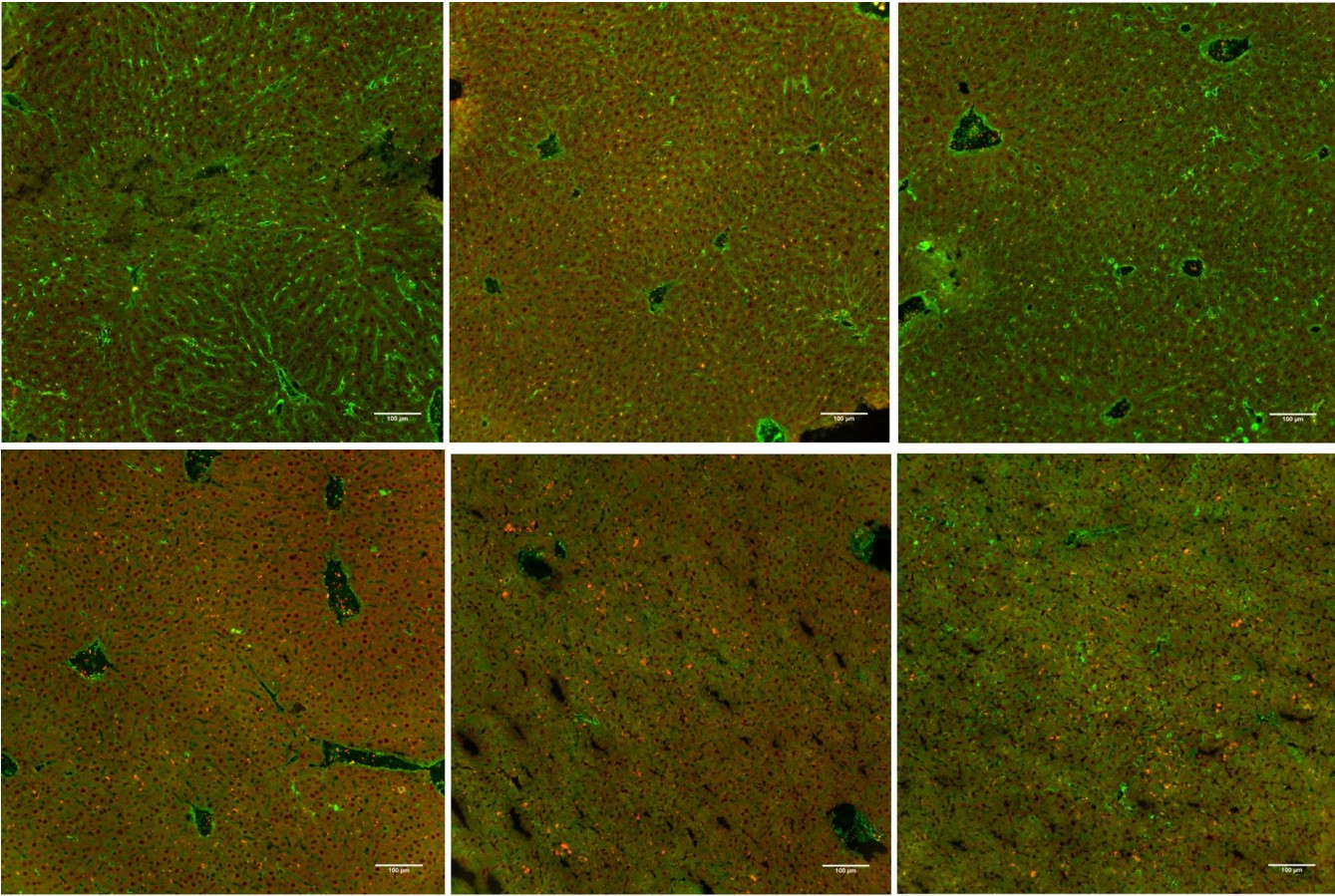

**Fig 6. Frozen sections of the liver with immunohistochemistry staining with an anti-fibrinogen antibody Alexa 488 and platelet anti-CD 42c antibody TRITC.** Pictured: left row untreated control animals (a-c) showing liver sections of untreated 8- to 10-week-old C57Bl/6 (3 examples a-c untreated and d-f CASP), the green dye (Alexa 488) represents the fibrinogen amount in the liver sinusoids, the orange dye (TRITC) shows platelet aggregates, the right column is 20 h after CASP surgery (d-f, scale bar 100 μm, negative control not shown).

transfer the ISTH 2001 criteria to the murine system for DIC research [13]. Admittedly, there are other criteria that are well known, e.g., the Japanese Ministry of Health and Welfare (JMHW) criteria and the Japanese Association for Acute Medicine (JAAM) criteria, each of which has advantages and disadvantages. We applied the ISTH criteria because they are well established [2, 18] and were previously used in other animal studies [13].

Differences from the human progression of sepsis have been observed in animal models. The fulminant sepsis of CASP-operated mice triggers hyperinflammation followed by immunosuppression. In the human system, however, the phase of hyperinflammation is often absent, and there is immunosuppression or hypoinflammation instead [19]. The different initial immunological situations of mice and humans may also produce variations in the frequency of thrombocytopenia. Only approximately 54% of human septic cases develop thrombocytopenia [20]. In our current paper, however, almost all CASP-operated mice developed a significantly lower platelet count after 20 hours (see Fig 1). It can be assumed that the thrombocytopenia of the mouse is proof of polymicrobial sepsis. The reasons for the high reproducibility of the results are the well-established CASP mouse model and the 16G CASP method used, which presents a model of severe sepsis with a mortality rate of 70% [21]. Hence, the first vital aspect of the ISTH score, thrombocytopenia, could be met in CASP. Clinical

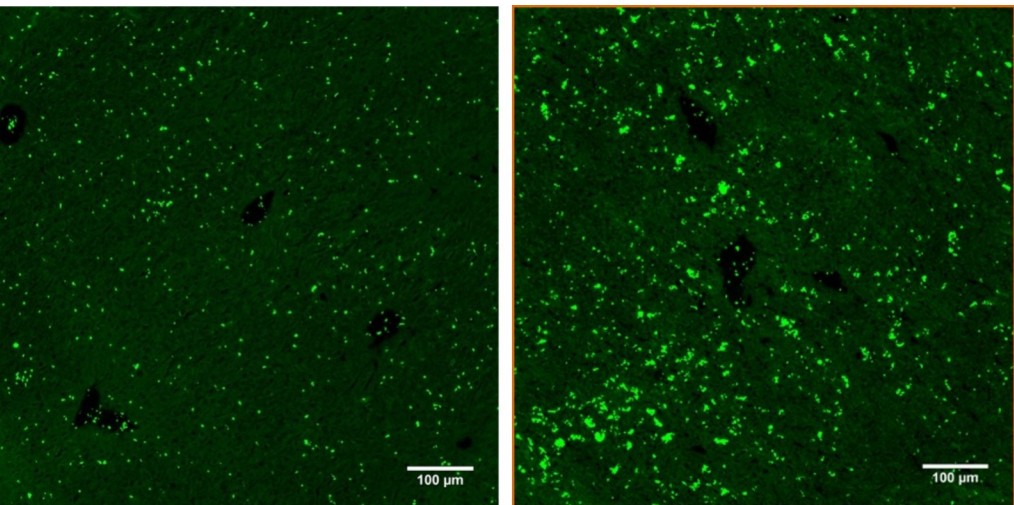

**Fig 7. Immunohistochemical staining of frozen liver sections 20 h after CASP with anti-CD42c\*FITC/Alexa 488.**
Depicted are representative results of frozen sections from 8- to 10-week-old C57Bl/6 liver tissue with immunohistochemistry staining of platelets (green dye) with anti-CD 42c antibody\*FITC/Alexa 488. The left row shows only a few platelets uniformly distributed in the liver tissue of untreated animals (a), while the right row shows platelet accumulation and aggregates in the liver tissue 20 h after CASP surgery (b); (scale bar 100 μm, negative control not shown, quantification of the r esults shown in Fig 8).

studies have shown that the extent of thrombocytopenia correlates with a poor survival rate of critically ill patients, thus being a negative predicative factor [22]. Various reasons for the decline of the platelets and its consequences for the organism have been frequently discussed and are as yet unanswered.

The current information is that thrombocytopenia (below 150,000 Plt/μl) may develop in humans through the involvement of platelets in the immune response to Gram-positive and Gram-negative bacteria. Platelets, in their role as covercytes, can directly interact with and eliminate bacteria and thus disappear from circulation [23]. An additional reason for the platelet decline may be the accumulation of platelets in the liver and spleen [24]. Severe sepsis may result in the already mentioned DIC and, by developing microthrombi in the atrial flow area, lead to multiple organ failure.

Due to thrombocytopenia being detected in the murine model, we were also interested in the question of whether thrombocytopenia influenced coagulation times. Owing to the fact that a healthy mouse has many more platelets in its circulation than a healthy human, it was questionable whether thrombocytopenia below 450,000 Plt/μl even had any influence on murine coagulation. This was, however, definitively answered by the prolonged bleeding time of the septic animals compared to the untreated control animals (see Fig 2).

An additional parameter for examining coagulation function is the mentioned prothrombin index in %, also referred to as the "Quick"-value. This value examines the extrinsic coagulation cascade. We analyzed the extrinsic coagulation cascade as an equivalent using rotational thromboelastometry (ROTEM®). To measure the coagulation time, we used thromboplastin with this test kit (tissue factor; ex-Tem®). As a result, we were able to show that the CASP-operated mouse had a longer extrinsic coagulation time (Fig 3). Admittedly, our data show the results of small groups (n = 6 untreated group; n = 5 CASP group) and present a large IQR of 409 s in the CASP-operated group, which is comparatively high relative to the IQR of 21 s in the untreated group. An explanation for the small group sizes is that the ROTEM®-Analyzer can test 4 probes in parallel, but since time is an important factor in analyzing blood clotting,

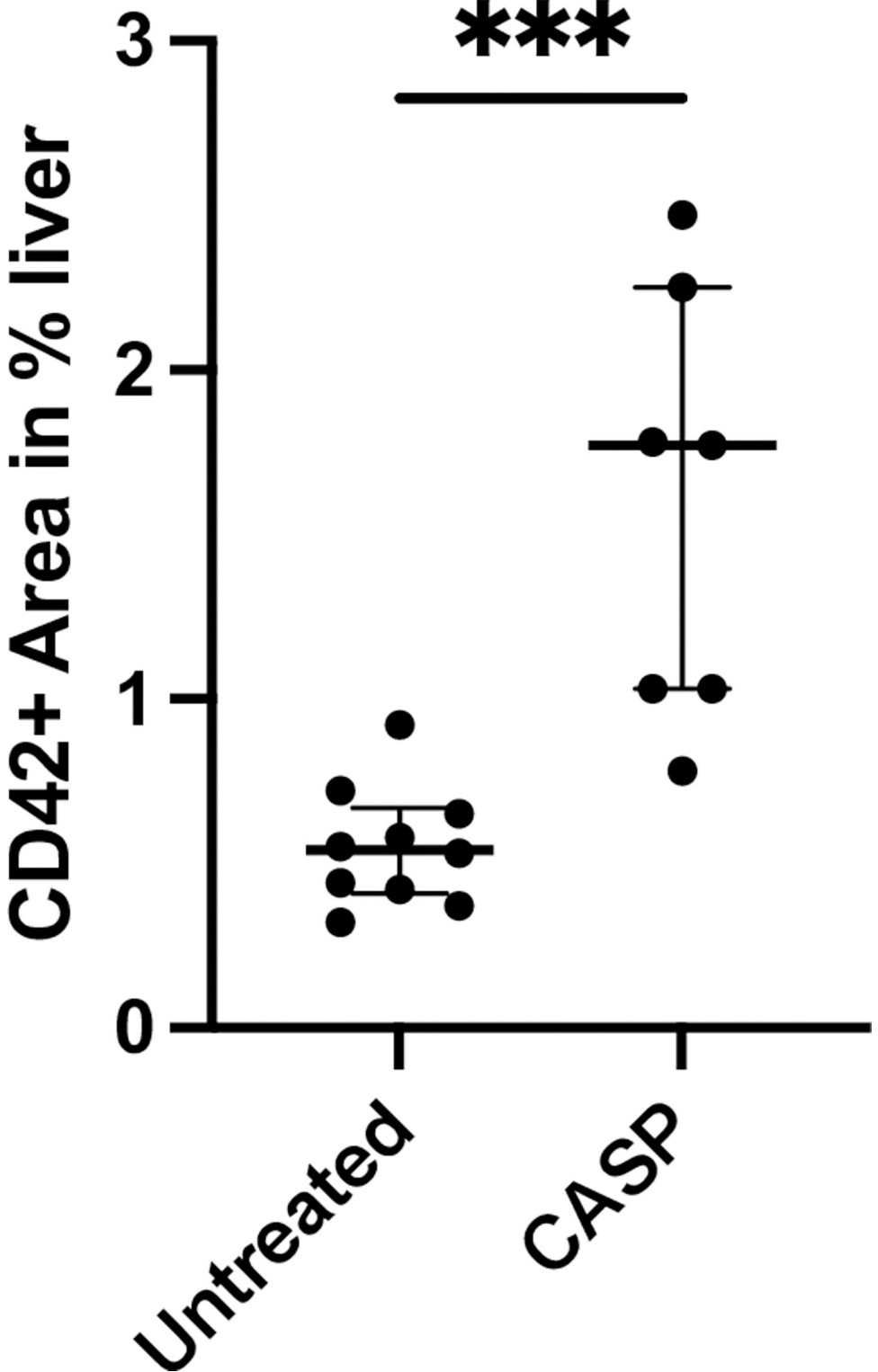

**Fig 8. Accumulation of hepatic platelets 20 hours after CASP surgery.** Pictured is the percentage area of FITC/Alexa 488-positive areas on the fluorescent section in proportion to the entire section. There was a significant increase (p < 0.001***) in the percentage of the FITC/Alexa 488-positive area, i.e., an increased accumulation of hepatic platelets in CASP-operated animals of 1.77% (1.69% IQR) of the area (n = 7) compared to untreated animals with 0.54% (0.6% IQR) of the FITC-positive area (n = 10). The bold black line depicts the median.

we minimized the time from taking blood samples to analysis; therefore, 2 experimental cycles were performed, and the tests were performed immediately after blood extraction. Due to the septic circulation conditions of the animals, it was not possible to obtain enough blood for multiple samples from any CASP-operated mouse.

In hospitals, thromboelastometry is an appropriate tool to rapidly obtain an overview of the performance of patients' intrinsic and extrinsic coagulation parameters. This method is of particular relevance in surgical transplantation and sepsis therapy [25]. Researchers were able to show that the 30-day survival probability was 85.7% among patients with normal ROTEM® parameters. If there was only one change in the measured coagulation parameters, the 30-day survival probability declined to 58.7% [26]. This observation is a strong indication of a lack of coagulation factors. Furthermore, additional criteria of the ISTH Consensus Conference 2001 could also be confirmed in our model. There were elevated D-dimer levels as well as decreased fibrinogen levels in the septic animals.

With regard to the severe thrombocytopenia in CASP, the question then was what had happened to the platelets missing from the circulation. To answer this question, we prepared frozen immunohistological sections of the liver, lung, and spleen. Our study was able to show that the liver parenchyma developed significant platelet accumulation 20 hours after CASP surgery (Figs 6A–6F and 7A, 7B). In doing so, the septic animals showed rather diffusely distributed platelet aggregations (Figs 6D–6F TRITC orange dye, 7b Alexa 488 green dye), while the untreated animals displayed regularly distributed, individual platelets in the parenchyma (Figs 6A–6C and 7A). There were, however, fewer platelets in the area of central veins (Fig 7B) in septic animals than in untreated control animals. An increase in the pooling of platelets in the liver of septic animals could also be verified by a significant increase in the fluorescent intensity after staining the platelets with CD42c*FITC/Alexa488 (Fig 7A and 7B). The pathological mechanisms of platelet accumulation are the subject of comprehensive analyses in the current literature. Aside from forming agglomerations, there are other explanations for their increased accumulation in the liver tissue. The literature also describes examinations of other models, such as the cecal ligation and puncture (CLP) and the LPS shock model, showing that platelets induce, among other effects, the recruitment of neutrophils into the vascular bed of the liver where they develop aggregates [27]. The sinusoids, situated behind these platelet-leukocyte aggregates, are then underperfused. The platelets also add to septic liver cell damage through interactions with neutrophil granulocytes [28]. A crucial factor for the accumulation of platelets in the capillary bed of the liver is the thrombocytic adhesion molecule CD62p. Other authors describe the accumulation of platelets as the body's defense mechanism against increased coagulopathy in the setting of DIC [29]. The hepatocyte asialoglycoprotein receptor, also referred to as the Ashwell receptor, was discovered in 2008 [30]. It is able to sequester platelets under the influence of a Gram-negative, systemic infection. For this purpose, bacterial neuraminidases help sequester the n-terminal acetyl lactosamine acids of glycoproteins on the host cells to allow their use in their own energy balance [31]. Bacteria that have neuraminidases in their repertoire of virulence factors and belong to the typical intestinal flora are, for example, *Bacteroides* spp. and *Escherichia coli*. Hence, the Ashwell receptor might also produce platelet aggregation in the liver of CASP models. However, the exact aggregation pathomechanism was not examined in this research paper.

The histological examinations could not identify a significantly increased aggregation of platelets in the spleen (S1 Fig). Human autopsies, too, only inconsistently reveal high microthrombosis in patients with a high DIC score [32, 33]. The reason for the difficult histological detection of microthrombosis might be the fact that DIC simultaneously causes hyperfibrinolysis, which might have already removed the thromboses that occurred. The chosen method of immunohistochemistry might not have been appropriate for our model to illustrate

sequestered platelets in the spleen. It might be that platelet sequestration by splenic macrophages triggered a reaction in the platelets, and the antibody used against CD42c could no longer be retained.

Under sepsis, neutrophil platelet aggregates may develop in the capillary flow area of the lung [34]. The aggregation of platelets in lungs during abdominal sepsis in humans as well as peritonitis in mice is well documented in the literature [35, 36]. The immunohistochemical analyses showed strong variations in the platelet aggregations in the alveolar flow area, thus preventing a conclusive statement (S2 Fig).

We were able to reconstruct DIC clinical parameters that met the definition of the ISTH 2001 Consensus Conference [7] in the CASP model. There was a significant decrease in the platelet count, the plasma fibrinogen level was significantly reduced, and the bleeding time was significantly prolonged. We were also able to show that this led to platelet aggregation in the liver parenchyma. D-dimer is an additional parameter of DIC. This fibrin degradation product gives information on plasmin activity and fibrinolysis. A study by the University of Zurich showed that an increase in D-dimer above the normal reference value occurred after abdominal surgical interventions. The maximum value was reached on the seventh postoperative day, with the values returning to the reference range only on the twenty-fifth postoperative day [37]. Since CASP is a surgical model for inducing severe sepsis, an elevation of D-dimer can be expected 20 hours after CASP surgery. Consequently, the elevated D-dimer level had no significance for assessing DIC in the model used.

Thus, we have been able to verify all ISTH 2001 parameters that define DIC, and from our point of view, the CASP model is an appropriate animal model to exemplify sepsis-induced DIC. As a result, DIC can be studied in this complex context, such as in terms of its medication responses, in a highly reproducible animal model.

## Conclusion

The CASP model is appropriate to analyze DIC diagnostic criteria. Currently, there are few appropriate DIC animal models with the majority being created by injecting substances, such as thromboplastin or lipopolysaccharide, to generate increased coagulation [13]. These models lack a realistic simulation of severe sepsis. The advantage of the CASP model is its more realistic simulation of polymicrobial sepsis following a major abdominal surgical intervention. Additionally, the model provides the opportunity to analyze the effect of new medications on septic organisms and to detect the influence of coagulopathy on outcomes. All in all the CASP model is suitable to analyze the complexity of DIC in polymicrobial sepsis.

## Supporting information

**S1 File.**
(ZIP)

**S1 Fig. None accumulation of platelets in spleen 20 hours after CASP surgery.** Immunohistochemical staining of frozen spleen sections 20 h after CASP with anti-CD42c*FITC/Alexa 488. Depicted are representative results of frozen sections from 8- to 10-week-old C57Bl/6 organ tissue with immunohistochemistry staining of platelets with anti-CD 42c antibody*FITC/Alexa 488. The sections show distribution of platelets in spleen tissue: untreated mice (left figure) and CASP operated (right figure) (scale bar in μm).
(TIF)

**S2 Fig. None accumulation of platelets in lung 20 hours after CASP surgery.** Immunohistochemical staining of frozen lung sections 20 h after CASP with anti-CD42c*FITC/Alexa 488.

Depicted are representative results of frozen sections from 8- to 10-week-old C57Bl/6 organ tissue with immunohistochemistry staining of platelets with anti-CD 42c antibody*FITC/Alexa 488. The sections show distribution of platelets in lung tissue: untreated mice (left figure) and CASP operated (right figure) (scale bar in μm).
(TIF)

## Acknowledgments

We would like to thank the Institute of Immunology and Transfusion Medicine, Department of Transfusion Medicine, University Medicine Greifswald for their assistance and opportunity to work with ROTEM® delta. Especially, I would like to express my gratitude to and respect for Birgitt Fürll. She was an inspiring and extraordinary human and a dedicated scientist, with a passion for platelets; unfortunately, she passed away.

## Author Contributions

**Conceptualization:** Julia van der Linde, Claus-Dieter Heidecke.

**Data curation:** Julia van der Linde, Thorben Klee.

**Formal analysis:** Julia van der Linde, Stephan Diedrich, Thorben Klee, Stephan Kersting, Wolfram Keßler.

**Investigation:** Julia van der Linde.

**Methodology:** Julia van der Linde.

**Project administration:** Claus-Dieter Heidecke.

**Validation:** Julia van der Linde, Stephan Diedrich, Wolfram Keßler.

**Visualization:** Julia van der Linde, Wolfram Keßler.

**Writing – original draft:** Julia van der Linde, Stephan Diedrich, Claus-Dieter Heidecke, Stephan Kersting, Wolfram Keßler.

**Writing – review & editing:** Julia van der Linde, Stephan Diedrich, Thorben Klee, Claus-Dieter Heidecke, Stephan Kersting, Wolfram Keßler.

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
