## [Decision Letter · Decision Letter 0]

18 Jul 2022

PONE-D-22-09810Disseminated Intravascular Coagulation (DIC): Old player creates new perspectives on the polymicrobial sepsis model of CASP.PLOS ONE

Dear Dr. van der Linde,

Thank you for submitting your manuscript to PLOS ONE. After careful consideration, we feel that it has merit but does not fully meet PLOS ONE’s publication criteria as it currently stands. Therefore, we invite you to submit a revised version of the manuscript that addresses the points raised during the review process.

Both reviewers felt the manuscript had value. There were, however, a few minor alterations requested. This includes refining the introduction to remove contradictory statements and a brief explanation of the inflammasome. A comment on the limited number of mice included should also be included.==============================

We look forward to receiving your revised manuscript.

Kind regards,

Elizabeth S. Mayne, M.D.

Academic Editor

PLOS ONE

Journal Requirements:

And finally please provide the timepoint at which the animals were sacrified

Reviewers' comments:

Reviewer's Responses to Questions

**Comments to the Author**

1. Is the manuscript technically sound, and do the data support the conclusions?

Reviewer #1: Partly

Reviewer #2: Yes

2. Has the statistical analysis been performed appropriately and rigorously? 

Reviewer #1: Yes

Reviewer #2: Yes

3. Have the authors made all data underlying the findings in their manuscript fully available?

Reviewer #1: No

Reviewer #2: Yes

4. Is the manuscript presented in an intelligible fashion and written in standard English?

Reviewer #1: Yes

Reviewer #2: Yes

5. Review Comments to the Author

Reviewer #1: This manuscript describes the coagulation profile of a mouse model of polymicrobial intra-abdominal sepsis (CASP), and demonstrates potential as a reliable animal model for further research on the pathogenesis and management of disseminated intravascular coagulation. It is overall well written, but requires some modification:

1) The data analysis has largely been performed on very small numbers of mice, and does not appear to be likely to be normally distributed. As such, the mean results reported may be skewed due to the impact of one or two outliers (particularly in the case of the ROTEM data and the bleeding times). Median and interquartile ranges should be reported instead of means and standard deviations. With these results, it will be easier to determine if all of the coagulation parameters are indeed significantly different in the CASP mice.

2) All of the experiments were not performed on all of the mice. This was explained in the discussion, but should also be mentioned in the methods, along with the overall number of mice tested (including the controls).

3) In the introduction, the inflammasome is referred to in line 72. A brief explanation of what the inflammasome is would be of value.

4) In line 82, it is stated that “Bacteremia is vital for developing DIC,”. This is not correct, as many non-infectious causes of DIC are known (as mentioned previously in the introduction).

5) In line 86, the meaning of the phrase “with a high yielding production” is unclear.

6) The manufacturer along with the manufacturers city and country should be specified for all kits and instruments used (including the Vetscan, the ROTEM, and the D-Dimer and Fibrinogen kits).

7) In line 128-130, the meaning of the following sentence: “Data from two experiments were pooled to minimize the time between blood sampling and analysis of the probes in ROTEM® delta.” is unclear.

8) For all the results, the number of included mice is given as a range (eg n=5-8 or n=23-25). Why is this? Please explain in the methods.

9) In line 245, “HE staining of the liver” is mentioned. What is this? Please include in the methods.

10) In line 254, the FITC positive area is reported. How is this calculated? Please include in the methods.

Minor grammatical corrections:

1) In line 16 it is stated that “In surgical ICUs, DIC is mainly caused by abdominal sepsis”. Please rephrase this as follows: “In surgical ICUs, DIC is frequently caused by abdominal sepsis”

2) In lines 45-47, the following: “The complexity and necessity of understanding coagulation disorders in the critically ill became highlighted during COVID-19 (3). This observation illustrates the significance of coagulation’s influence on the prognosis of critically ill patients.” Should be rephrased as follows: “The complexity and necessity of understanding coagulation disorders in the critically ill was highlighted during the COVID-19 pandemic (3), where the influence of coagulation abnormalities on the prognosis of critically ill patients was illustrated.”

3) In line 50, I suggest removing “micro-organisms” from the list of causes of DIC.

4) In line 58, please change “increase the intake of platelets” to “increase the consumption of platelets”.

5) The sentence in lines 258-261 should be rephrased as follows: “The double staining of fibrinogen and platelets in CASP sepsis clearly verified increased fibrinogen in the area of the sinusoids in untreated mice, while fibrinogen was less readily seen in those with sepsis.”

Reviewer #2: With this manuscript, the authors showed that the CASP model is suitable for analysing the complexity of DIC in polymicrobial sepsis. The experiments were performed on 8-10 weeks old female C57BL/6 mice, but the number of animals used was not mentioned in the methodology section. Treated and non-treated animals were compared in terms of platelet count, bleeding time, TEG, D-Dimer, Fibrinogen and immunohistological staining for platelets and fibrinogen. Again, the number of blood samples used for these tests were not mentioned.

The authors found that the CASP model caused thrombocytopenia, elongation of bleeding time, reduction in fibrinogen levels and the presence of D-Dimers in plasma. They mentioned that the CASP model seems superior to other models for inducing DIC, however they did not compare the different models directly. They should rather stated that the CASP model is suitable to analyse the complexity of DIC in polymicrobial sepsis.

This paper is furthermore well-written and recommended for publication.

6. PLOS authors have the option to publish the peer review history of their article (what does this mean?). If published, this will include your full peer review and any attached files.

Reviewer #1: No

Reviewer #2: **Yes: **Muriel Meiring

---

## [Author Response · Author response to Decision Letter 0]

26 Sep 2022

Dear reviewer no1

Thank you for your comments, we appreciate your effort to us and have endeavored to adapt the charts and present them with the median values and interquartile measurements accordingly. We added every single note conscientiously.

Thank you for your advice: The different group sizes are due to the complex methods used. On the one hand, the animals in the CASP-operated group might had a Stress Severity Score of over 10 and where released from the experiments prematurely, on the other hand, in our experience blood sampling is more difficult in septic animals, this fact was taken into account when choosing a larger group size of CASP.

The normal coagulation parameters of C57Bl/6 are well known, so that a large control group could be dispensed with, provided that the experimental method itself does not pose large variables.

We changed the ranges and explained it separately.

Dear reviewer no2

Thank you for your statement and comments, we completed in line 590

“All in all CASP model is suitable to analyze the complexity of DIC in polymicrobial sepsis.”

Best wishes to all

kind regards 

Julia van der Linde

---

## [Decision Letter · Decision Letter 1]

28 Oct 2022

Disseminated Intravascular Coagulation (DIC): Old player creates new perspectives on the polymicrobial sepsis model of CASP.

PONE-D-22-09810R1

Dear Dr. van der Linde,

We’re pleased to inform you that your manuscript has been judged scientifically suitable for publication and will be formally accepted for publication once it meets all outstanding technical requirements.

Kind regards,

Elizabeth S. Mayne, M.D.

Academic Editor

PLOS ONE

Additional Editor Comments (optional):

Reviewers' comments:

Reviewer's Responses to Questions

**Comments to the Author**

1. If the authors have adequately addressed your comments raised in a previous round of review and you feel that this manuscript is now acceptable for publication, you may indicate that here to bypass the “Comments to the Author” section, enter your conflict of interest statement in the “Confidential to Editor” section, and submit your "Accept" recommendation.

Reviewer #1: (No Response)

Reviewer #2: All comments have been addressed

2. Is the manuscript technically sound, and do the data support the conclusions?

Reviewer #1: Yes

Reviewer #2: Yes

3. Has the statistical analysis been performed appropriately and rigorously? 

Reviewer #1: Yes

Reviewer #2: Yes

4. Have the authors made all data underlying the findings in their manuscript fully available?

Reviewer #1: No

Reviewer #2: Yes

5. Is the manuscript presented in an intelligible fashion and written in standard English?

Reviewer #1: Yes

Reviewer #2: Yes

6. Review Comments to the Author

Reviewer #1: Thank you for your changes. The manuscript requires very minor grammatical corrections, which I have made in the attached word document.

Reviewer #2: The changes recommended by the reviewers have been made. I recommend the manuscript for publication.

7. PLOS authors have the option to publish the peer review history of their article (what does this mean?). If published, this will include your full peer review and any attached files.

Reviewer #1: No

Reviewer #2: No

---

## [Editor Report · Acceptance letter]

29 Nov 2022

PONE-D-22-09810R1 

Disseminated Intravascular Coagulation (DIC): Old player creates new perspectives on the polymicrobial sepsis model of CASP. 

Dear Dr. van der Linde:

I'm pleased to inform you that your manuscript has been deemed suitable for publication in PLOS ONE. Congratulations! Your manuscript is now with our production department. 

Kind regards, 

on behalf of

Dr. Elizabeth S. Mayne 

Academic Editor

PLOS ONE